# Interaction effect: Are you doing the right thing?

Sara Garofalo [ID]*, Sara Giovagnoli, Matteo Orsoni, Francesca Starita [ID], Mariagrazia Benassi [ID]

Department of Psychology "Renzo Canestrari", University of Bologna, Cesena, Italy

* sara.garofalo@unibo.it

**Data Availability Statement:** Data and code are available at https://osf.io/ya9mp/.

**Funding:** The author(s) received no specific funding for this work.

## Abstract

How to correctly interpret interaction effects has been largely discussed in scientific literature. Nevertheless, misinterpretations are still frequently observed, and neuroscience is not exempt from this trend. We reviewed 645 papers published from 2019 to 2020 and found that, in the 93.2% of studies reporting a statistically significant interaction effect (N = 221), post-hoc pairwise comparisons were the designated method adopted to interpret its results. Given the widespread use of this approach, we aim to: (1) highlight its limitations and how it can lead to misinterpretations of the interaction effect; (2) discuss more effective and powerful ways to correctly interpret interaction effects, including both explorative and model selection procedures. The paper provides practical examples and freely accessible online materials to reproduce all analyses.

## Introduction

Methodological and statistical misinterpretations of the interaction effect are no news to many fields of science [1–10]. Since the 1970s, scientists and statisticians advised about the complicated nature of the interaction effect in order to both warn about possible inaccuracies in its interpretation and indicate best practices [7, 11]. Nevertheless, incorrect uses are still observed in literature and neuroscience is not exempt from this trend [12–14].

To get an overview of the practices currently most adopted to investigate interaction effects, the neuroscientific field served as a case study for the purposes of this paper, although the considerations here reported extend to all scientific fields. We reviewed all the articles pertaining to behavioral, cognitive, cellular, and molecular neuroscience published between 2019 and 2020 in two of the most prestigious journals in neuroscience, namely Neuron (N = 398) and Nature Neuroscience (N = 247) (Table 1). When a statistically significant interaction effect was reported (N = 221), we checked the strategy that was adopted to interpret its results (Fig 1). In 93.2% (N = 206) of cases, some form of pairwise comparison was involved; while only in 6.8% of cases (N = 15), the interaction effect was interpreted based on a descriptive interpretation of the means (i.e., without having to resort to a further inferential statistic—like a t-test—but simply describing the results using summary statistics—like mean and standard deviation or standard error). Amongst pairwise-comparisons, in a striking majority of cases (98.06%, N = 204), post-hoc contrasts between all factor levels were used, while, in a small minority of cases

**Competing interests:** The authors have declared that no competing interests exist.

(1.94%, N = 4), the number of comparisons was a-priori defined based on the experimental hypothesis. Detailed information on all reviewed papers is available online (see the "open materials" section). Based on these results, it appears that post-hoc pairwise comparisons are the designated method to interpret interaction effects, at least in the neuroscientific literature. This result calls for the need to further highlight the methodological and statistical drawbacks and implications of this approach, and provide clear and simple guidelines for the correct interpretation of the interaction effect, in line with previous evidence [1, 2, 4, 7–10, 12, 13].

In the neuroscientific literature, full factorial experimental designs (i.e., a design with two or more independent variables in which all the main and interaction effects are estimated) are often used aiming to obtain a statistically significant interaction effect. Groups and conditions

**Table 1. Number of studies reviewed reporting different approaches to the interpretation of interaction effects.**

| | | | Neuron | Nature Neuroscience |
|---|---|---|---|---|
| Interaction effect | Pairwise comparisons | Post-hoc | 154 | 48 |
| | | A priori | 2 | 2 |
| | Descriptive interpretation of means | | 9 | 6 |
| No interaction effect | | | 233 | 191 |
| Tot | | | 398 | 247 |

Note: Studies including only fMRI analysis were not included because of the fundamentally different principles and implications of the use of multiple comparison corrections that apply to that case (e.g., familywise error correction for clusters).

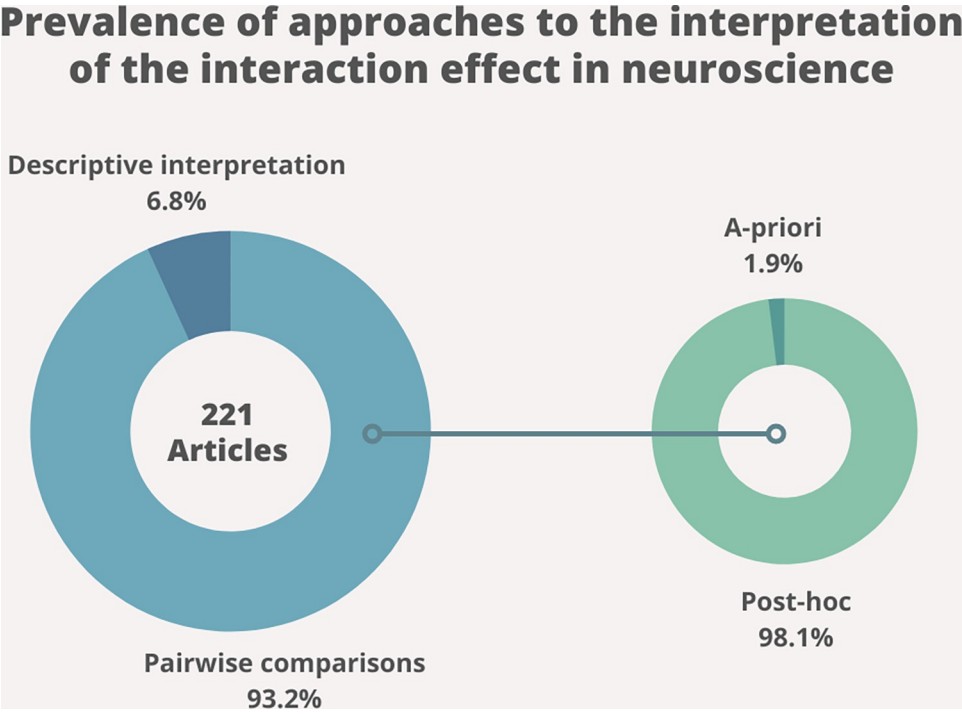

**Fig 1. Reviewed prevalence of approaches to the interpretation of interaction effects.** The blue plot on the left represents the percentage of approaches used to follow up a statistically significant interaction (N = 221), namely pairwise comparison (N = 206) or descriptive interpretation of the means (N = 15). The green plot on the right further specifies the type of pairwise comparison adopted, namely post-hoc comparisons between all factor levels (N = 204) or a-priori-defined comparisons (N = 4).

are carefully chosen to uncover precise expectations about their neurophysiological or behavioral expressions [2, 12]. Let's consider the following example: a researcher aims to test differences in arousal to the presentation of fearful and neutral stimuli between a group of participants with a selective lesion to the amygdala and a healthy control group. Based on previous evidence, a lesion of the amygdala can be expected to be associated lower level of arousal when presented with fearful (vs neutral) stimuli, as compared to a healthy control group [15–19]. This research hypothesis typically translated to a 2 (group: lesion/control) x 2 (stimulus: fearful/neutral) factorial experimental design.

Statistically speaking, in a factorial experimental design, interaction effects can be observed if the impact of one factor changes based on the levels of another factor [1]. If the variables meet the assumptions for parametric analyses, the statistical evaluation of a factorial design can be achieved via an analysis of variance (Anova), which separates the portion of effect that can be specifically attributed to each factor (main effect) from that representing their interdependence (interaction effect).

A frequent mistake when using this approach, is the tendency to read interaction effects even where there are none [12]. For example, a difference between fearful and neutral stimuli in the control group and the absence of such difference in the lesion group tends to be interpreted as evidence for a difference between the two groups [12]. Critically, this kind of conclusion is often reported even in absence of a statistically significant interaction or a direct comparison between the effect sizes of the contrasts between the two groups [12]. This erroneous interpretation can be found both in null hypothesis significance testing (NHST) and Bayesian perspective [13]. Indeed, even within this framework, a higher probability associated with the alternative hypothesis in one group ($BF_{10} > 1$) but not in another ($BF_{10} < 1$) is often erroneously interpreted as evidence of a difference between the two groups [13].

A more classical debate, however, concerns the mistakes characterizing the interpretation of a correctly reported statistically significant (typically intended as $p < 0.05$) interaction effect [1, 2, 4, 7–10]. Although the limitations of commonly used procedures, such as post-hoc contrasts, have been extensively recognized and discussed in the literature [1, 2, 4, 7–10], many neuroscientific studies still fail to take these indications into account. We argue that the reason behind this incorrect approach to interaction effect is twofold. First, the interaction effect represents a global difference between factors, which hardly fits into the typical neuroscientific experimental logic, characterized by direct comparisons of experimental groups or conditions, or subtractive methods. Thus, rather than taking a more global perspective, it is classically used to test very specific differences—i.e., hypotheses—between groups or conditions. Second, previous literature addressing how to interpret the interaction effect tended to adopt technical terminologies and methods that may not be easily accessible without a strong statistical background. Crucially, practical examples and guidelines on how to correctly perform and interpret interactions with currently available statistical tools are missing.

Our review indicates, at least in the neuroscientific field, a tendency to an oversimplification of the interaction effect based on the comparison between all factorial sub-groups/sub-conditions on a two-by-two basis (i.e., pairwise comparisons). While this approach may seem easier than looking at the big picture arising from the interaction effect, it poses a series of methodological and statistical pitfalls that actually complicate or mislead the interpretation of the results [2, 20, 21].

Given the widespread use of post-hoc pairwise comparisons, the present paper aims to: (1) highlight the limitations of this approach and how it can lead to misinterpretations of the interaction effect; (2) show how and why a more informative explorative interpretation of interaction effects can be conveyed by a descriptive and visual inspection of the model estimated marginal means and errors; and (3) highlight that, when an interaction effect is

designed to uncover a specific hypothesis, model selection procedures (such as Bayesian informative hypotheses) can be a more appropriate tool to provide a sensible answer to the initial research question. All examples and analyses reported in the paper can be reproduced using the data, code, and step-by-step guidelines available online (see the "open materials" section).

## The use and misuse of post-hoc pairwise comparisons

Post-hoc pairwise comparisons consist of contrasting, on a two-by-two basis, all the levels contained within the factors involved in a statistically significant interaction. Considering the 2 (group: lesion/controls) x 2 (stimuli: fearful/neutral) design of our example, the interaction effect can be followed up by a series of pairwise comparisons between lesion-neutral vs lesion-fearful, control-neutral vs control-fearful, and so on (up to 6 comparisons). For each comparison, in the NHST approach, the probability associated with the data given a true null hypothesis (p-value) is obtained, and the interpretation proceeds based on this evidence.

This approach presents several drawbacks that will be described in the following paragraphs.

### A global effect cannot be reduced to pairwise contrasts

Probably, the major drawback of pairwise comparisons is that they represent an oversimplification of the interaction effect. By separately looking at the difference between pairs of sub-groups/sub-conditions, they fail to represent the effect as a whole. Interaction effects should be intended as evidence of a global reciprocal influence among the factors, which can hardly be described by independently contrasting two levels at a time. As mentioned before, the sole evidence that the two groups present comparable arousal levels for neutral stimuli but not for fearful ones is not sufficient to represent an interaction effect [12]. Similarly, evidence that the lesioned group shows no difference in arousal to fearful vs neutral stimuli, while the control group does, is not sufficient either. To provide a complete interpretation, it is essential to observe how values are modulated by all levels of the factors simultaneously.

While the presence of a statistically significant interaction effect accounts for all these changes at once, contrasting pairs of levels ignores (one at a time) one crucial comparison conceived as a control condition for the initial hypothesis (e.g., if we focus on comparing the control and lesion group in the fearful condition, we are ignoring the possibility of a difference between groups also in a neutral control condition) [2], thus disregarding the interaction as a global effect.

### Correcting for multiple comparisons can lead to false positives and false negatives

The second drawback of pairwise contrasts concerns the need for multiple comparison corrections. When performing several comparisons (i.e., between all possible pairs of groups or conditions), the probability to find that one of these is statistically significant just by chance (i.e., false positives, Type I error) increases drastically and needs to be controlled with a more stringent threshold for statistical significance (α-level). Amongst other limitations, the debated mathematical adjustments of the α-level proposed in the literature (e.g., Bonferroni, Tukey, etc.) [22–25] inevitably result in increasing other sources of error, like the chance of false negatives (Type II error). Moreover, Type III (i.e., correctly rejecting the null hypothesis for the wrong reason) and Type IV errors (i.e., incorrect interpretation of a correctly rejected hypothesis) are around the corner [2, 26–28].

In our review, several types of corrections for multiple comparisons were reported (Fig 2). More specifically, 70 studies used Bonferroni's correction, 53 studies used Tukey's correction,

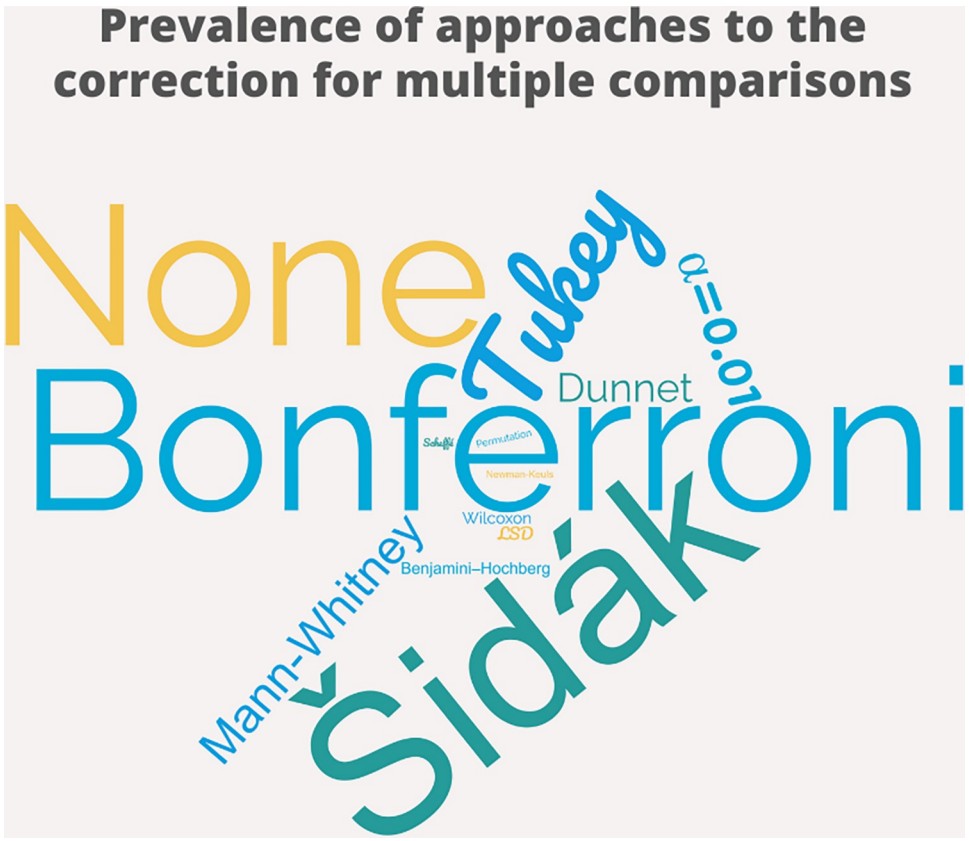

**Fig 2. Reviewed prevalence of approaches to the correction for multiple comparisons.** This word-cloud plot represents the frequency of the reported approaches to multiple comparison correction. Font size is scaled by a factor proportional to its frequency so that the bigger, the more frequent. Bonferroni (N = 70), Tukey (N = 53), and Šidák's (N = 52) corrections were the most adopted. 36 studies reported no correction for multiple comparisons. In all cases, no clear justification for the use of one, multiple, or no type of multiple comparisons correction was provided.

52 studies used Šidák's correction, 2 studies used LSD corrections, and 1 study used permutations [22, 23, 25, 29]. 17 of these papers reported a combination of these or Dunnett, LSD, Mann-Whitney, Newman-Keuls, Scheffè, and Benjamini–Hochberg's corrections. 7 studies used an arbitrary threshold of $\alpha = 0.001$ or $\alpha = 0.01$ was used. 36 studies used no correction for multiple comparisons. In all cases, no clear justification for the use of one, multiple, or no type of multiple comparisons correction was provided. Of note, almost all pairwise comparisons reported (98.06%, N = 204) were based on post-hoc comparisons, i.e. all possible pairs of sub-levels were contrasted against each other. In only 4 studies (1.94%) the number of comparisons performed was defined a-priori, based on the experimental hypothesis, although sometimes erroneously referred to as planned contrasts [30].

## Absence of evidence is not evidence of absence

A further problem with post-hoc pairwise comparisons is that, even if we do find a statistically significant difference in the experimental group but not in the control group, we cannot conclude that there is no effect in the latter [13, 20, 31, 32]. Nevertheless, the initial hypothesis is often based on this implicit assumption. Following the NHST approach, a $p > 0.05$ does not allow us to conclude that there is no difference between the two conditions. The absence of evidence for a difference between conditions is not evidence of the absence of difference between

conditions. When such a crucial piece of information for our hypothesis—e.g., the absence of effect in the control group—is taken for granted only because $p > 0.05$, the resulting conclusion is necessarily flawed from a statistical point of view [13, 20, 31–33].

## Redundant statistical tests

Another limitation of the use of pairwise comparisons is their redundancy. The presence of a statistically significant interaction among factors is already tested within the Anova model. The use of additional statistical tests (i.e., evaluation of a p-value for each pair of contrasts) is redundant, as the interaction effect alone (when significant) is sufficient to indicate that there is a different trend among the levels of each—or some of the—factors [1, 7].

Once a statistically significant interaction among factors is established, the next step is to move from testing (i.e., binary interpretation of the presence/absence of the effect) to estimation (i.e. obtaining an accurate valuation of the parameters and their uncertainty) without unnecessary additional statistical tests [34–37].

## Observed vs estimated marginal means

Although reporting observed (raw) means and errors (e.g., standard deviation, standard error) is always desirable, these are not appropriate for interpreting interaction effects emerging from a statistical model that accounts and corrects for the variability associated with all factors considered. As such, observed means and errors cannot represent the statistical interaction resulting from the Anova [1, 2, 7, 38].

Classically, the type of Anova traditionally used (the standard in most GUI-based statistical software is type III sum of squares) assumes that the total variance of the model is given by the variance of each individual factor (main effect) plus the variance represented by the interdependence between the factors (interaction effect) [21, 39, 40]. To disentangle the main effects from interactions, each effect is evaluated after removing all the others. Thus, each effect is estimated based on the residual variance that can be specifically ascribed to it, cutting away the variance attributed to other main or interaction effects. Model estimated marginal means and errors, on the other hand, represent the estimation of each effect "cleansed" of the variability due to other factors included in the statistical model [1, 2, 7, 21, 38]. In contrast, observed (raw) means and errors contain the interaction effect as well as the main effect of each individual factor [1, 2, 7, 38].

More specifically, while the observed and estimated *means* may coincide if the factorial design is balanced (i.e., an equal number of subjects or observations per condition), when the design is unbalanced or when a covariate or a third factor have a significant impact on the dependent variable, also the estimated means drastically change as compared to the observed ones [41]. Critically, the observed and model-estimated *errors* are always different. This is a crucial piece of information for the inferential conclusions of a post-hoc analysis because the result of a t-test is based on a comparison of the errors and, thus, how these are calculated plays a big role in the final result [42].

Based on these considerations, it should become evident how using independently calculated t-tests to follow up a statistically significant interaction is inherently wrong. Nevertheless, this remains a common incorrect practice. If pairwise comparisons are necessary, they should always be based on estimated marginal means and errors, as they represent the parameters net of the other effects controlled within the initial Anova model.

Unfortunately, the terminology around this issue is usually vague in research reports, making it hard to evaluate whether actual post-hoc comparisons (i.e., based on model estimated marginal means) or separate t-test/Anova comparisons (i.e., independently tested on observed

means) were performed. Pairwise comparisons are generally referred to as "post-hoc" whether they are based (estimated) or not based (observed) on the original statistical model, simply because they are used to follow-up a previously obtained (i.e., "post") statistically significant interaction effect. But this terminology is misleading.

In our review, 19 studies reported using pairwise comparisons via t-tests, but there was no sufficient information to determine whether these were computed on observed or estimated marginal means and errors. Although disregarded in most papers, this information is crucial to understand whether the variability and the error terms reported in the interaction effect and the follow-up comparisons stem from the same statistical model [1, 2, 7, 21, 38].

## A more appropriate way: Describe interaction effects via estimated marginal means and confidence intervals

As previously stated, a statistically significant interaction should be intended as a global effect, indicative of the presence of different trends among all the sub-group/sub-conditions involved, in which each factor is evaluated net of the influence of all other factors (i.e., Anova with type III sum of squares). Looking at a polished estimation of the interaction effect is especially important when control groups/conditions or covariates are present, as it takes out the amount of variability that is not attributed to the group/condition of interest. For example, one may not be simply interested in whether patients respond differently than controls, but in whether such difference can be specifically attributed to the fearful stimulus, net of all other factors. Of note, although the present paper focuses on interaction effects, the same logic applies to a correct interpretation of main effects.

Many authors converge on the idea that the most effective way to represent and interpret interaction effects goes through a simple descriptive interpretation of the means and errors estimated based on the Anova model [1, 2, 7, 36, 43–45]. Notably, in our review, only 6.8% (N = 15) of studies adopted a descriptive interpretation of the interaction effect, although it was not possible to clarify whether such interpretation was based on observed means or estimated marginal means and errors.

A way to provide an effective descriptive interpretation of the interaction effect based on meand and means and 95% confidence intervals (CIs) is illustrated in Fig 3. The characteristics

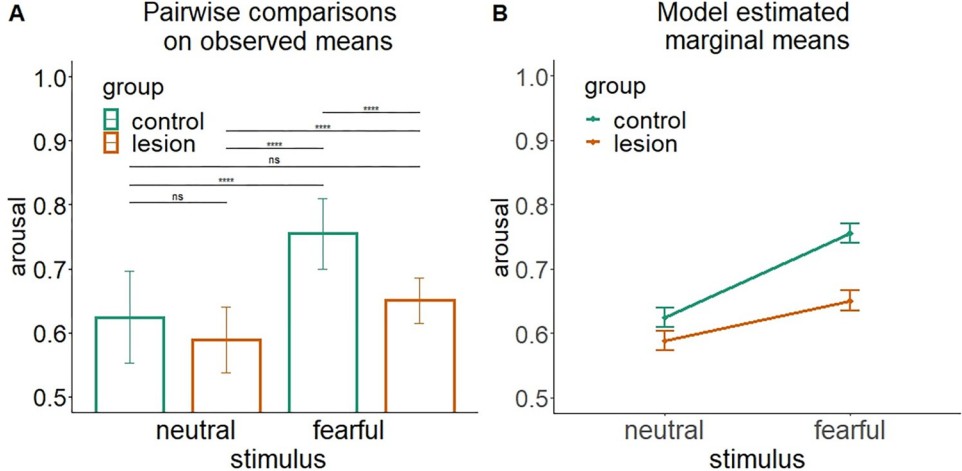

**Fig 3. Two ways to represent and interpret an interaction effect arising from a 2 (group: lesion/control) x 2 (stimuli: fearful/neutral) design.** (A) Barplot with observed means (± standard error) as classically used to represent post-hoc pairwise comparisons between all sublevels involved in an interaction effect (**** = p<0.001, ns = p>0.05). (B) Model estimated marginal means and 95% confidence intervals.

and advantages of using CIs for this purpose are illustrated in Box 1. The example in Fig 3 represents data from 100 simulated cases representing a 2 (group: lesion/control) x 2 (stimuli: fearful/neutral) between-subjects two-way Anova. The results show the presence of a statistically significant main effect of stimulus type ($F_{(1, 196)} = 140.82$, $p < 0.001$), group ($F_{(1, 196)} = 81.125$, $p < 0.001$), and group by stimulus interaction ($F_{(1, 196)} = 19.48$, $p < 0.001$). Fig 3A shows observed means and standard errors as typically represented with a barplot, reporting above the result of Bonferroni-corrected pairwise t-tests on observed means, calculated to

## Box 1. Confidence Intervals

Confidence intervals indicate the precision of an estimate [68, 69] and are based on specific assumptions about the statistical model (e.g., normally distributed values in the population). They provide (1) a point estimate, that is the value that most likely represents the population parameter based on N hypothetical replications of the sample data, and (2) an interval which represents the precision of such estimate, i.e. how likely it is that it contains an error (the narrower, the better) [36, 43, 44]. The values within the confidence intervals can be interpreted as a range of plausible values in the population, while the values outside the interval as a range of implausible values in the population [43, 70]. Hence, this information can be used to describe the overall trend emerging from the ineraction effect and the strength of the difference between the sub-groups/sub-conditions based on how much the intervals overlap.

One of the main advantages of confidence intervals is that they encourage a more accurate interpretation of results in terms of quantity ("how strong is the difference across conditions?") and direction of the observed pattern ("in what way they differ?") [35, 55, 71, 72].

Moreover, they provide information on a measurement scale that makes sense for the research question as are reported in the same measurement unit of the variable of interest (i.e., seconds, number of responses, voltage, etc.) [36, 43–45]. Differences can be quantified in terms of number of responses, milliseconds, or voltage, thus allowing the attribution of a practical meaning to the variations observed across groups or conditions [36, 43–45]. Whether such difference (or lack of) is "big enough", depends on the nature of the variables and the research question at stake. For instance, a 2 sec difference between two groups in a simple reaction time task (i.e., press the button when any stimulus appears) is intuitively more meaningful than if the same difference is observed in a complex choice reaction time task (e.g., press the button when you achieve the result of this equation).

It is important to note that a comparison of CIs based on the extent of their overlap is correct only if means are independent, which should typically be the case for between-subjects designs [44]. If there is little ($< 25\%$ of the full CI length) or no overlap, there is reasonable evidence of a difference between the two population means [44]. In the case of repeated measures, the CIs of the paired differences can be calculated. In this paper, all the examples are based on between-subjects designs assuming independence between the means [44, 52].

follow up the statistically significant interaction effect. In contrast, Fig 3B shows model estimated marginal means and 95% confidence intervals.

Comparing the two plots in Fig 3, it becomes immediately clear how the same data can give rise to distinct interpretations based on how they are reported. The results presented in Fig 3A lead to the conclusion that the two groups present significantly different arousal levels when it comes to fearful stimuli ($p<0.001$) but not for neutral stimuli (ns, $p>0.05$) [12]. However, when looking at Fig 3B, such a conclusion drastically changes. Although both groups show higher arousal to fearful than neutral stimuli and this effect is stronger in the control group than in the lesioned group, there are critical group differences that could call for a different interpretation of the results. Namely, the control group presents higher levels of arousal relative to the lesioned group, regardless of the type of stimulus. This result suggests that the different psychophysiological patterns observed may not be specifically attributed to a difference in processing fearful stimuli, but rather ascribed to a more general reduced arousal level characterizing lesioned patients. Of note, the data reported in Table 2 show how, in this example, although the observed and estimated marginal means coincide, their standard error does not (see the previous paragraph "observed vs estimated marginal means" for the implications of such difference).

In a second example (Fig 4), we used a new dataset representing the same model and added a third factor (e.g., working memory ability) that has a significant impact on the dependent variable but no role in the interaction effect. In other words, the Anova model presents a statistically significant main effect of stimulus type ($F_{(1, 192)} = 5.25$, $p = 0.02$), group ($F_{(1, 192)} = 8.57$, $p = 0.003$), working memory ($F_{(1, 192)} = 68.3$, $p < 0.0001$), and group by stimulus interaction ($F_{(1, 196)} = 4.02$, $p = 0.04$) but no interaction of the working memory with any other factor ($ps>.86$). In such a scenario, despite affecting the overall model (main effect), the working memory would typically be ignored in the investigation of the group by stimulus interaction. However, as clearly evident in Fig 4 and Table 3, not only the errors but also the means change based on whether we are looking at the observed or the model estimated ones. As a consequence, the interpretation of the results may also drastically change.

Both examples provided in this section can be replicated in R [46, 47] and Jamovi [48] following the code and step-by-step guidelines available online (see the "open materials" section).

## Post-hoc pairwise comparisons lead to higher false positive risk than estimation based on confidence intervals

To support the method here proposed, we compared the false positive risk (i.e., the risk to reject the null hypothesis when it is true) [49, 50] emerging from the pairwise comparisons based on post-hoc t-tests performed on the observed means versus the overlap between 95% CIs of the estimated marginal means [51].

To this purpose, we performed 5000 simulations of 2x2, 2x3, and 3x3 between-subjects Anova models. The null hypothesis was posed as true by looking at differences between

**Table 2. Observed vs estimated means and standard errors.**

| group | stimulus | observed mean | SE | estimated marginal mean | SE |
|---|---|---|---|---|---|
| control | neutral | 0.624 | 0.010 | 0.624 | 0.008 |
| lesion | neutral | 0.589 | 0.007 | 0.589 | 0.008 |
| control | fearful | 0.755 | 0.008 | 0.755 | 0.008 |
| lesion | fearful | 0.651 | 0.005 | 0.651 | 0.008 |

The data refer to the example reported in Fig 3

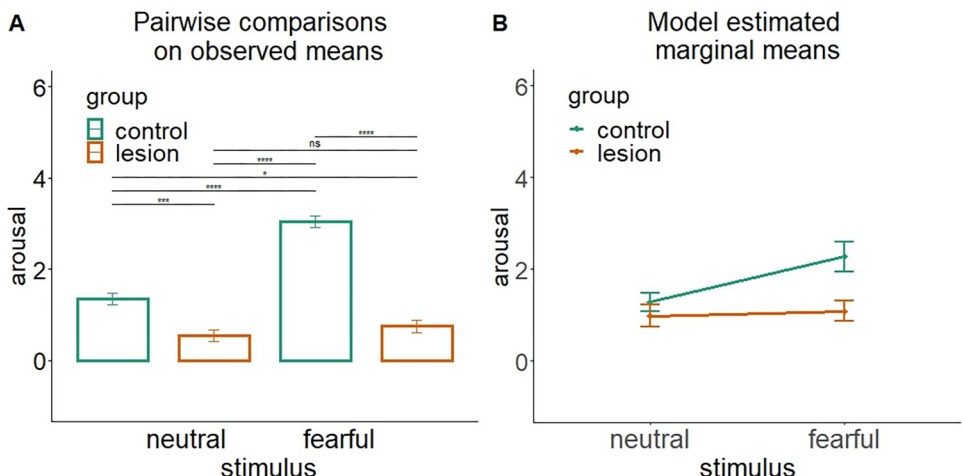

**Fig 4. Two ways to represent and interpret an interaction effect arising from a three-way Anova having as independent variables: Group (lesion/control), stimulus (fearful/neutral) and working memory ability.** The plot represents the statistically significant group by stimulus interaction. The working memory did not significantly interact with any variable. (A) Barplot with observed means (± standard error) as classically used to represent post-hoc pairwise comparisons between all sublevels involved in an interaction effect (**** = p<0.001, ns = p>0.05). (B) Model estimated marginal means and 95% confidence intervals.

observations that have comparable true means, and a standard deviation of 1. The power was modulated by considering a variable number of subjects per group (N = 5, 10, 20, 30, 50, or 100) and a variable average difference between the group means (average mean difference = 0, 0.1, 0.25, or 0.5). For each simulation, when the model resulted in a statistically significant interaction between the two factors (~5% of times), we counted the number of times in which (a) Bonferroni-corrected pairwise comparisons based on observed means and errors were statistically significant (p<0.05) and (b) 95% confidence intervals of the estimated marginal means overlapped for less than 25% of the full CI length. Of note, the overlap treshold for CIs was based on previous literature [44, 51, 52] to find a criterion comparable with a p<0.05. The false positive risk was then calculated as the relative frequency (number of instances over the total number of comparisons) of cases meeting this criterion.

The results (Fig 5) show that for all Anova models (2x2, 2x3, and 3x3), number of subjects per group (N = 5, 10, 20, 30, 50, or 100), and average difference between means (0, 0.1, 0.25, or 0.5), post-hoc pairwise comparisons systematically presented a higher false positive risk, thus indicating that the use of post-hoc pairwise comparisons based on observed means and errors increases the probability that the observed result occurred by chance only, as compared to an interpretation based on estimated marginal means and CI [49, 50].

**Table 3. Observed vs estimated means and standard errors.**

| group | stimulus | observed mean | SE | estimated marginal mean | SE |
|---|---|---|---|---|---|
| control | neutral | 1.346 | 0.128 | 1.281 | 0.100 |
| lesion | neutral | 0.539 | 0.132 | 0.981 | 0.120 |
| control | fearful | 3.034 | 0.131 | 2.274 | 0.167 |
| lesion | fearful | 0.746 | 0.140 | 1.086 | 0.111 |

The data refer to the example reported in Fig 4

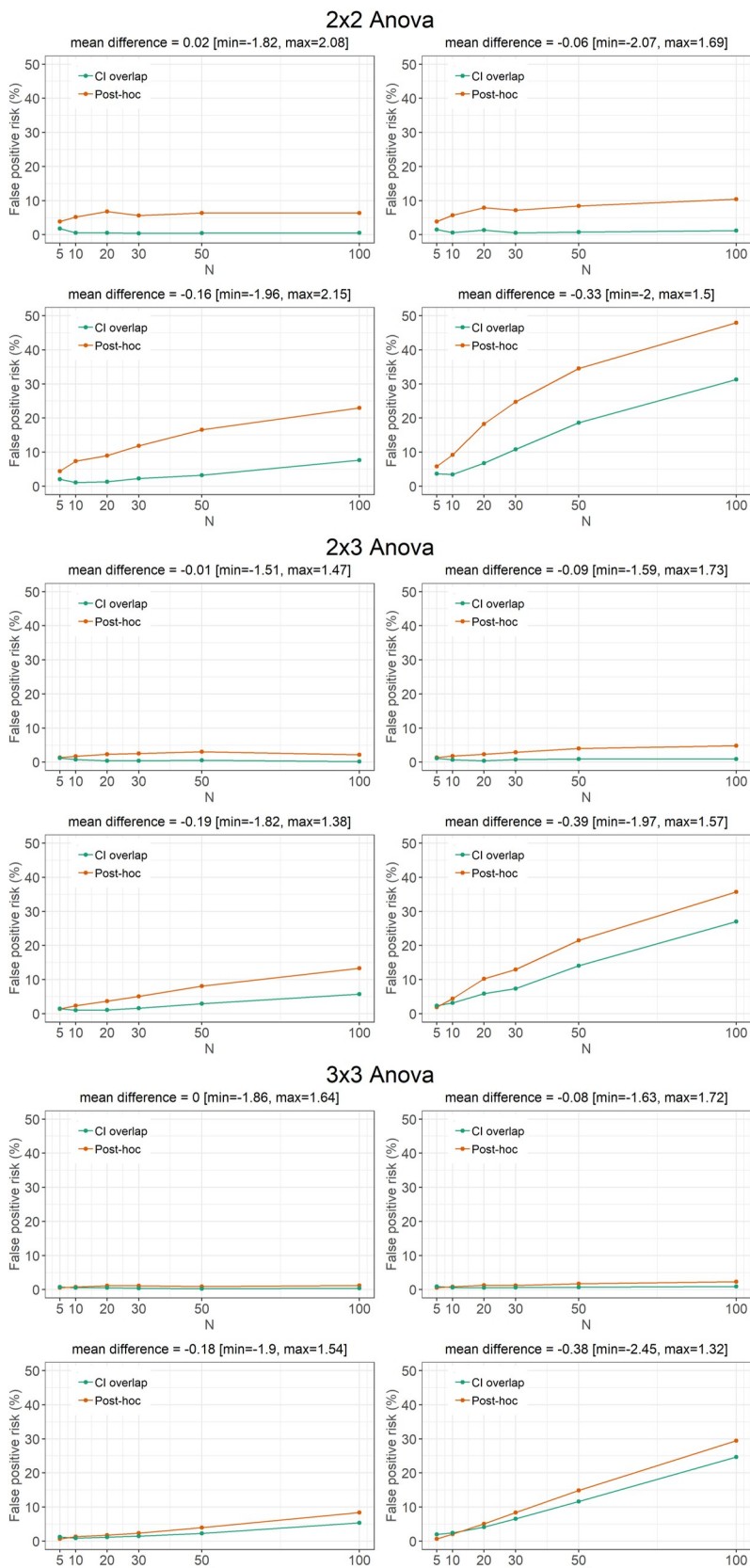

**Fig 5. Results of 5000 simulations of two-ways Anova with different number of levels (2x2, 2x3, and 3x3), number of subjects per group (N = 5, 10, 20, 30, 50, or 100), and average difference between means (0, 0.1, 0.25, or 0.5).** The null hypothesis was posed as true by looking at differences between observations which have comparable true means, and a standard deviation of 1. The false positive risk represents the relative frequency (%) of either (in orange) Bonferroni-corrected pairwise comparisons based on observed means and errors were statistically significant (p<0.05) or (in green) 95% confidence intervals of the estimated marginal means overlapping for less than 25% of the full CI length.

The simulation can be replicated in R [46, 47] following the code available online (see the "open materials" section).

## Testing specific hypotheses

The approaches described above are considered explorative and can be useful when no specific hypotheses are present. However, factorial experimental designs are often implemented to test very precise predictions. For instance, in our example, the 2x2 factorial design is set to reveal an interaction effect between the group (lesion/control) and the stimulus type (fearful/neutral) to support the expectation that patients with a lesion to the amygdala will show lower arousal when presented with a fearful, as compared to a neutral, stimulus, relative to the control group; but also that the two groups shall be comparable when facing a neutral stimulus. Such specific research hypotheses are not best answered based on the sole presence of main or interaction effects as offered by multivariate Anova. A direct contrast between competing hypotheses via model selection procedures can be much more informative [31, 53, 54].

### Planned constrasts

Although poorly used, planned comparisons provide a limited but valid approach to compare simple ordering of means that could represent competing hypotheses [30]. The first characteristic of planned comparisons is that they are based on the idea that only orthogonal comparisons need to be planned in advance, based on the researcher's expectations. If a hypothesis exists, comparing all sub-groups/sub-conditions with each other—as done with post-hoc pairwise comparisons–is unnecessary as only comparisons reflecting the actual experimental hypothesis should be tested. This approach better controls for Type I and Type II errors. The second characteristic is that planned comparisons can be performed between one sub-group/sub-condition and all others, thus not limiting the interpretation to a contrast between two levels (like in pairwise comparisons).

Nevertheless, orthogonal planned comparisons still present most of the limitations described above for pairwise comparisons, namely failing to take into account the interaction as a global effect, the need to correct for multiple comparisons, and the use of redundant statistical tests. Moreover, only simple relations between a single parameter against all the others can be included in each hypothesis. For instance, we could compare the hypothesis that the lesion-neutral condition presents lower arousal than all other conditions (i.e., lesion-fearful, control-neutral, control-fearful) or that the lesion-fearful condition presents lower arousal than all other conditions (i.e., lesion- neutral, control-neutral, control-fearful), but this would not be very informative in respect to the initial research hypothesis [20, 31, 32, 53, 55–60].

### Bayesian informative hypotheses

Bayesian informative hypotheses offer an effective alternative to overcome the limitations of pairwise comparisons and planned contrast to directly compare specific research hypotheses within a Bayesian inferential framework (Box 2) [32, 53, 54, 61].

## Box 2. Bayesian inference

The most commonly adopted Bayesian approach to inference is to compare the posterior probability of the alternative hypothesis ($H_1$, difference between groups or conditions) with the posterior probability of the null hypothesis ($H_0$, absence of difference between groups or conditions) via a Bayes Factor (i.e., $BF_{10}$) [73–75]. As compared to null hypothesis significance testing (NHST), this approach has the merit to shift the focus onto the hypothesis we aim to test ($H_1$) and overcome dualistic reasoning by quantifying and comparing the evidence reflected in the data for the two hypotheses. Nevertheless, the actual experimental hypothesis is not necessarily well represented by a classically defined alternative hypothesis, in which the difference between two groups or conditions is simply expected to be higher or lower than zero. Further problems are represented by the search for (often inadequate) rules of interpretations of the Bayes Factor values—which fall back to dualistic inference—and the absence of proper use of prior distributions [32, 76].

Informative hypotheses are defined as hypotheses formulated to reflect research expectations in terms of inequality constraints amongst parameters [31, 53, 61]. These constitute a more powerful tool than planned comparisons, as a partial ordering of means (i.e., including only some levels) or specific mathematical relationships between parameters (e.g., differences, products) can be used to represent experimental hypotheses [53]. For example, if a smaller difference in arousal is expected between neutral and fearful stimuli in the lesioned group, as compared to the control group (i.e., an interaction effect as previously intended), this hypothesis can be represented as follows:

$$H_1 = (\mu_{lesion-fearful} - \mu_{lesion-neutral}) < (\mu_{controls-fearful} - \mu_{controls-neutral})$$

Crucially, other hypotheses may also be relevant to the researcher, such as controlling for the possibility that the fearful stimuli elicit a higher arousal regardless of group:

$$H_2 = (\mu_{lesion-fearful}, \ \mu_{controls-fearful}) > (\mu_{lesion-neutral}, \ \mu_{controls-neutral})$$

or that lesioned participants present systematically lower arousal than controls but modulated by the type of stimulus:

$$H_3 = \mu_{lesion-neutral} < \mu_{lesion-fearful} < \mu_{controls-neutral} < \mu_{controls-fearful}$$

or that lesioned participants present systematically lower arousal than controls regardless the type of stimulus:

$$H_4 = \mu_{lesion-neutral} < \mu_{lesion-fearful} < \mu_{controls-neutral} < \mu_{controls-fearful}$$

The contrast between models can also include an unconstrained hypothesis (Hu), which is a hypothesis representing all possible sets of relationships between the parameters without constraints ($H_u = \mu_{lesion-neutral}, \mu_{lesion-fearful}, \mu_{controls-neutral}, \mu_{controls-fearful}$). The formulation of a model representing the null hypothesis is not mandatory and, as for any other model, should only be included if meaningful from a scientific point of view [31, 53].

Bayesian informative hypotheses allow us to compare and contrast such a set of predefined hypotheses via a model selection procedure in which each hypothesis (or model) represents a

possible explanation of the phenomenon. For each hypothesis, the posterior model probability (PMP) is calculated via the Bayes theorem and expressed with a value between 0 and 1. This value can be interpreted as the relative amount of support for each hypothesis given the data and the set of competing hypotheses included (the sum of all posterior model probabilities adds up to 1). The model with the highest PMP reflects the best hypothesis, i.e. the hypothesis with the highest relative probability [31, 53, 54, 62, 63]. To further support model selection, the PMPs can also be compared via Bayes Factor to (a) that of the other hypotheses tested, (b) to its complement hypothesis (i.e., a model that contains any set of restrictions between the parameters except the one represented by the hypothesis tested), or (c) to the unconstrained hypothesis (Hu) [31, 53, 63].

When testing these hypotheses on the data from the previous example, the results showed that the second model ($H_2$) is associated with the highest relative posterior model probability (Fig 6). This indicates that the hypothesis that fearful stimuli elicit higher arousal regardless of the group is more likely than that of a selective difference between groups over fearful stimuli. Of note, $H_2$ basically describes the main effect of stimulus that, although statistically significant in the Anova model, was disregarded based on the presence of the interaction effect. By comparing the probability associated with alternative explanations, it is possible to observe how that hypothesis is the most likely, given the data.

A full description of this approach is beyond the purposes of this paper; for an exhaustive overview of this approach and tutorials, see Béland et al., 2012; Hoijtink, 2012; Hoijtink et al., 2019a. The example here provided can be replicated in R [46, 47] and Jasp [64] following the code and step-by-step guidelines available online (see the "open materials" section).

## Conclusion

In the age of the replication crisis, it is crucial to acknowledge that the data analysis tools available today offer many ways to easily overcome bad practices of the past, even if commonly adopted, and embrace a more accurate interpretation of results.

We suggest that, when a factorial experimental design is used, how to explore a resulting interaction effect deserves careful consideration. Despite being the most diffuse approach (Table 1), post-hoc pairwise comparisons present several shortcomings, like failure to acknowledge the interaction as a global effect, the need for multiple comparisons correction, the impossibility to test the absence of difference, and the use of a redundant statistical evaluation. This calls for clear advice on the need to identify more appropriate strategies to correctly interpret interaction effects. A general overview of the strengths and weaknesses of the approaches reviewed in the present paper is provided in Table 4, along with practical indications of the research scenarios in which their use is appropriate or to be avoided.

In general, when no specific research expectations are present, in line with previous literature [1, 2, 4, 7–10], post-hoc pairwise comparisons can be effectively replaced with the descriptive and visual inspection of the model estimated marginal means and 95% confidence intervals [45, 52, 65–67]. Confidence intervals offer an immediate visual representation of the results based on a measurement scale that makes sense for the research question. As such, they can be used for the interpretation of the interaction effect without the need for further and redundant statistical tests. Crucially, this approach shifts the focus on how the values of experimental factors are modulated by all levels of the factors, not just some (or pairs) of them. This is, by definition, what the interaction effect actually is: a complex net of relationships between all the factors and their sublevels. Nevertheless, factorial designs and interaction effects are neither the best nor the only statistical approach available. When a specific research hypothesis is present, planned contrasts offer a viable, although limited, solution to represent and contrast

# Posterior Model Probabilities

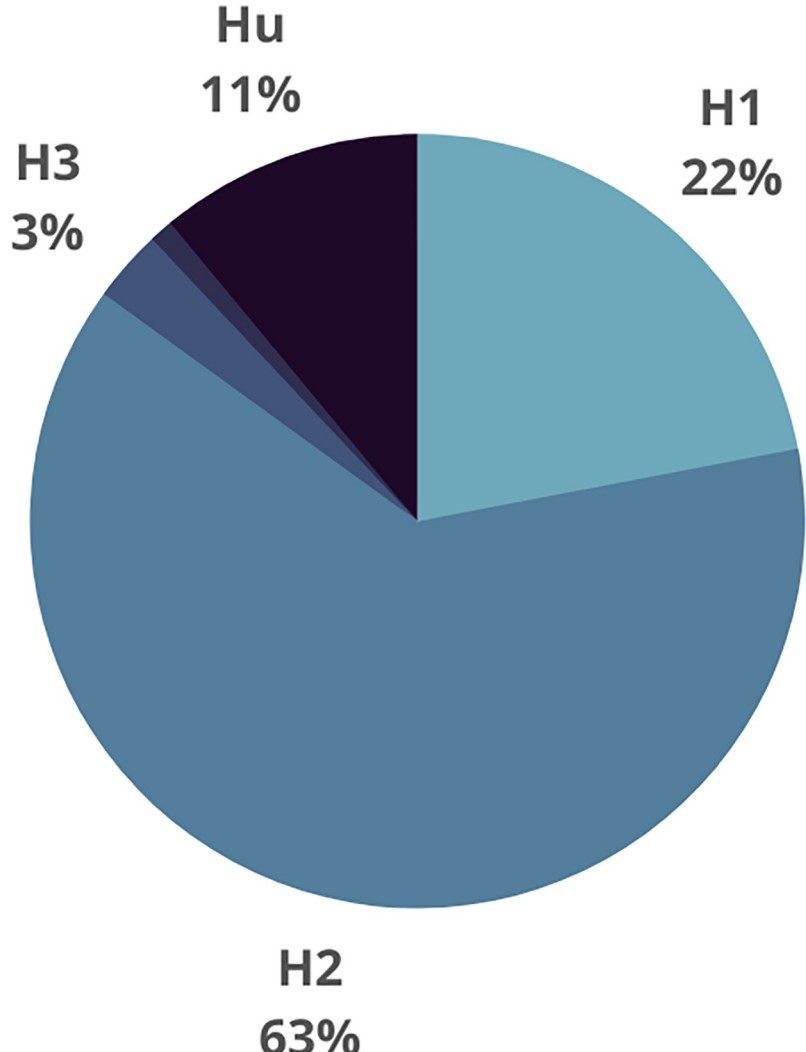

**Fig 6. Posterior model probabilities of the three models (H$_1$, H$_2$, and H$_3$) were estimated via Bayesian informative hypotheses.**

some forms of research expectations. Crucially, model selection procedures such as Bayesian informative hypotheses [31, 53, 54], can more powerfully test precise research hypotheses by enabling the definition of all relevant hypotheses in terms of inequality constraints among parameters and the comparison of their associated probability within a Bayesian inferential framework [31, 53, 61].

## Open materials

All examples and analysis reported in the present paper can be reproduced using the data, code, and step-by-step guidelines available at https://osf.io/ya9mp/.

**Table 4. Overview of the characteristics and experimental scenarios in which the approaches to the investigation of the interaction effect discussed din the present paper should or should not be used.**

| | Post-hoc pairwise comparison using observed means and errors | Post-hoc pairwise comparisons using estimated marginal means and errors | Descriptive interpretation of estimated marginal means and 95% CIs | Planned contrasts | Bayesian informative hypotheses |
|---|---|---|---|---|---|
| Can answer a specific hypothesis | no | no | yes | yes | yes |
| Can be used for an explorative approach | yes | yes | yes | yes | yes |
| Parameters are estimated based on the tested model | no | yes | yes | yes | yes |
| Does not require correction for multiple comparisons | no | no | yes | no | yes |
| Can be used to describe absence of difference | no | no | yes | no | yes |
| Any combination of comparisons is possible | yes | yes | yes | no | yes* |

* but not ideal, comparisons should only be included if reflecting actual research hypothesis

## Acknowledgments

We thank Luigi A. E. Degni and Daniela Dalbagno for their invaluable help in collecting and reviewing the articles for this paper.

## Author Contributions

**Conceptualization:** Sara Garofalo, Mariagrazia Benassi.

**Data curation:** Sara Garofalo, Sara Giovagnoli, Matteo Orsoni, Mariagrazia Benassi.

**Formal analysis:** Sara Garofalo.

**Methodology:** Sara Garofalo.

**Software:** Sara Garofalo.

**Supervision:** Sara Giovagnoli, Mariagrazia Benassi.

**Validation:** Francesca Starita, Mariagrazia Benassi.

**Visualization:** Francesca Starita, Mariagrazia Benassi.

**Writing – original draft:** Sara Garofalo, Mariagrazia Benassi.

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
