## [Decision Letter · Decision Letter 0]

31 Mar 2022

PONE-D-22-05978Interaction effect: do the right thing.PLOS ONE

Dear Dr. Garofalo,

Thank you for submitting your manuscript to PLOS ONE. After careful consideration, we feel that it has merit but does not fully meet PLOS ONE’s publication criteria as it currently stands. Therefore, we invite you to submit a revised version of the manuscript that addresses the points raised during the review process. Both reviewers liked your manuscript and believed that it could be helpful to researchers. One reviewer noted that the primary topic of your manuscript was interaction effects, but you presented no new information about them specifically, and that makes your paper similar to a review paper. Review papers are outside PLOS ONE's scope.

I'm inclined to agree with that reviewer and suggest that you consider his or her suggested changes and add new material about interactions. In particular, it would be helpful to PLOS ONE readers if you compared the methods rather than simply stating them.

Thank you for your hard work--your motiving neuroscience example is a good one.

We look forward to receiving your revised manuscript.

Kind regards,

Richard Evans

Academic Editor

PLOS ONE

Journal Requirements:

Reviewers' comments:

Reviewer's Responses to Questions

**Comments to the Author**

1. Is the manuscript technically sound, and do the data support the conclusions?

Reviewer #1: Partly

Reviewer #2: Partly

2. Has the statistical analysis been performed appropriately and rigorously? 

Reviewer #1: N/A

Reviewer #2: Yes

3. Have the authors made all data underlying the findings in their manuscript fully available?

Reviewer #1: Yes

Reviewer #2: Yes

4. Is the manuscript presented in an intelligible fashion and written in standard English?

Reviewer #1: Yes

Reviewer #2: Yes

5. Review Comments to the Author

Reviewer #1: In this paper the authors argue that neuroscientists tend to do the wrong thing when interpreting and building on significant interaction effects. I am generally sympathetic towards papers like this, that aim to educate researchers with regard to statistical issues. I also believe that the present paper has the potential to make a useful contribution in this regard. However, as I will explain below there are quite a number of arguments in the current draft that to me are confusing or seem wrong. I would like to see these issues addressed in a potential revision.

1) About the section ‘A global effect cannot be reduced to pairwise effects’. I agree that an interaction effect cannot be reduced to a number of pairwise comparisons. However, there are a few sentences in the argument here that I do not understand:

- “The two groups may present a different level of arousal regardless of the type of stimulus.” I don’t see why this is an argument; there may be a significant interaction effect AND a significant main effect of group. These are not mutually exclusive.

- “Within each group, the difference may be very small.” Please clarify.

- “…, contrasting pairs of levels ignores (one at a time) one crucial comparison conceived as control

condition for the initial hypothesis [2]”. What do the authors mean with ‘the initial hypothesis’? Please rephrase this sentence.

2) About the section ‘Observed vs estimated marginal means’. It is important that the authors make clear that observed means and estimated means are only different from each other in unbalanced factorial designs, and explain what it means that a statistical design is balanced or unbalanced. I would think that many neuroscience studies use balanced designs.

3) Throughout the paper the authors refer to ‘a descriptive interpretation of the means’, but it never became fully clear to me what this means. It would help a lot if the authors could provide a few examples (i.e. bit of text) of descriptive interpretation of the means, e.g. applied to their toy data.

4a) Figure 3 is presented to serve as an important illustration of the authors’ point about the use of comparisons on observed versus marginal means. However, to my surprise the model-estimated marginal means (right panel) in this figure seemed to be identical to the observed means (left panel). I do not understand why the authors chose an example where the model-estimated means are identical to the observed means, when they want to emphasize that one is more informative than the other.

I also have several questions about the text on page 14 that accompanies Figure 3.

4b) First, the authors say that compared to Figure 3a, Figure 3b makes it way more clear that “…the control group presents higher levels of arousal relative to the lesioned group, regardless of the type of stimulus.” I do not agree. To me, the bar plot and line plot are equally clear in terms of illustrating possible main effects and interactions. We are talking about only 2 x 2 data points and the simple use of color and location (left vs right) to discriminate the 4 conditions. Why would one plot be more clear than the other?

4c) Second, the authors write that “This result suggests that the different psychophysiological patterns observed should not be specifically attributed to a difference in processing fearful stimuli, but rather ascribed to a more general reduced arousal level characterizing lesioned patients.”

Again (see my point 1 above) the authors imply that there can be EITHER an interaction effect OR a main effect. Why can’t there be an interaction effect AND a main effect? That is, the patients show a generally reduced arousal response compared to the controls (main effect) but more so for the fearful than for the neutral stimuli (interaction effect)? I find this quite confusing.

4d) Altogether, to me, the only essential difference between Figures 3A and 3B (with regard to the topic of the present paper) is that a bar plot makes it more easy to illustrate the significance (ns vs ***) of pairwise comparisons than the line plot. Is this perhaps the main point that the authors want to make about the choice between bar plots or line plots?

5) “Of note, H2 basically describes the main effect of stimulus that, although statistically significant in the Anova model, was disregarded based on the presence of the interaction effect.” This is yet another statement suggesting that according to the authors you either interpret the interaction effect or the main effect. I believe that this is based on a fundamental misunderstanding. I refer the authors to this webpage: https://www.theanalysisfactor.com/interpret-main-effects-interaction/

** Minor comments **

There are several places in the paper (including the abstract) where the authors write ‘revised’/’revising’ but actually mean ‘reviewed’/’reviewing’

There are a few places in the paper (including the abstract) where the authors write ‘remark’ but actually mean ‘discuss’

The figure captions of Figure 1 and Figure 2 contain a lot of details that are copied more or less from the main text. This redundancy is undesirable.

Page 10. “Another limitation of the use of pairwise comparisons is its redundancy.” Should ‘its’ be ‘their’?

Page 10. “The use of additional statistical tests (i.e., evaluation of a p-value for each pair of contrasts) is redundant, as that alone is sufficient to indicate …”. It is unclear what ‘that’ refers to.

Page 13. “data from 100 simulated cases representing a a 2 (group: lesion/control) x 2 (stimuli: fearful/neutral) design are reported” >> a a is typo

Page 19. “By comparing the probably associated with alternative explanations …” >> probability

“Bayesian Informative Hypotheses” >> do not capitalize informative and hypotheses

Figure 3. Typo ‘marignal means’

Reviewer #2: General comments

This manuscript will provide an essential reference for researchers trying to "do the right thing" when assessing interactions.

1. The authors should consider a different title. The reason is that the article lists several alternative statistical methods, but it doesn't give a single method to "do the right thing." In other words, the title is a little misleading.

2. I'm concerned that the article fails to meet publication criteria for PLOS ONE. It appears to be primarily a review article describing different methods of analyzing interaction effects. PLOS ONE typically does not publish review articles. I understand that your neuroscience example is original material, but the manuscript primarily discusses interaction effects.

My suggestion is to add some original material about interactions, possibly a comparison among the methods with data or simulated data. I would also suggest giving PLOS ONE readers more guidance on when to use the specific methods or recommend one method over the others.

6. PLOS authors have the option to publish the peer review history of their article (what does this mean?). If published, this will include your full peer review and any attached files.

Reviewer #1: No

Reviewer #2: No

---

## [Author Response · Author response to Decision Letter 0]

30 Jun 2022

Please refer to the Cover Letter uploaded and labeled 'Response to Reviewers', which contains a point-by-point answer to all reviewers' comments

---

## [Editor Report · Decision Letter 1]

6 Jul 2022

Interaction effect: are you doing the right thing?

PONE-D-22-05978R1

Dear Dr. Garofalo,

We’re pleased to inform you that your manuscript has been judged scientifically suitable for publication and will be formally accepted for publication once it meets all outstanding technical requirements.

Kind regards,

Richard Evans

Academic Editor

PLOS ONE

Additional Editor Comments (optional):

Thank you for working hard to respond to the reviewers' comments.
---

## [Editor Report · Acceptance letter]

7 Jul 2022

PONE-D-22-05978R1 

Interaction effect: are you doing the right thing? 

Dear Dr. Garofalo:

I'm pleased to inform you that your manuscript has been deemed suitable for publication in PLOS ONE. Congratulations! Your manuscript is now with our production department. 

Kind regards, 

on behalf of

Dr. Richard Evans 

Academic Editor

PLOS ONE